# Cavity-QED-controlled two-dimensional Moiré excitons without twisting

Francesco Troisi [1] ✉, Hannes Hübener [1] ✉, Angel Rubio [1,2] ✉ & Simone Latini [1,3] ✉

We propose an all-optical Moiré-like exciton confinement by means of spatially periodic optical cavities. Such periodic photonic structures can control the material properties by coupling the matter excitations to the confined photons and their quantum fluctuations. We develop a low energy non-perturbative quantum electro-dynamical description of strongly coupled excitons and photons at finite momentum transfer. We find that in the classical limit of a laser driven cavity the induced optical confinement directly emulates Moiré physics. In a dark cavity instead, the sole presence of quantum fluctuations of light generates a sizable renormalization of the excitonic bands and effective mass. We attribute these effects to long-range cavity-mediated exciton-exciton interactions which can only be captured in a non-perturbative treatment. With these findings we propose spatially structured cavities as a promising avenue for cavity material engineering.

Two-dimensional van der Waals materials have emerged as a versatile platform for opto-electronic devices thanks to their highly tunable properties, such as optical and electronic band gaps and dielectric response[1–5]. Among this class of materials, transition metal dichalcogenides (TMDs) have attracted particular interest due to their strongly bound excitons and related optical activity in the visible range[6–8]. Excitons do indeed dominate the dynamics of several phenomena of TMDs, such as valley polarization and non-linear optical response[9,10]. Hence, controlling the behavior of excitons offers a direct route to program optical functionalities in this class of materials. Twist engineering[11–13], a technique in which atomically thin TMD layers are stacked with a predefined twist angle, allows for spatial control of excitons and consequent modification of their optical activity[14]. This is possible because twisting two-dimensional (2D) crystals with respect to each other leads to the formation of Moiré patterns, which generate spatially periodic electrostatic potentials strong enough to influence the motion and confinement of the otherwise free excitons, c.f. Fig. 1a. Aside from excitonic control, twist engineering, also dubbed twistronics, allows for generating novel quantum phases[5,15], including correlated insulator states[16], unconventional superconductivity[12], and fractional Chern insulators[17].

In this work, we propose an alternative strategy for spatial confinement of excitons that utilizes optical cavities instead of twist engineering. Optical cavities confine the electromagnetic field in a small volume, strengthening its intensity and making it possible to strongly couple to excitons of embedded TMDs. Such interaction can in turn lead to the formation of exciton-polaritons[18–22]. Previous works show that due to the strong light-matter interaction, planar optical cavities allow the control of the creation and mixing of polaritons made of composite excitons[18], as well as excitons, phonon and photon quasi-particles, the so-called phonoriton[23]. Cavity confinement not only alters the excitonic spectrum[24,25] but can also give rise to new equilibrium quantum phases due to the coupling of the host material with the vacuum fluctuations of light[26–33]. Here, we propose using spatially structured optical cavities, such as those sketched in Fig. 1, to produce an all-optical Moiré-like exciton confinement without twisting. This is a different paradigm compared to previous works that deal with exciton-polaritons in grated systems as they aim to reshape the properties of emitted light[20,34]. With our approach, instead, we demonstrate that we can control the matter properties by specifically structuring the excitonic quasi-particle, using both quantum fluctuations and classical fields.

[1]Max Planck Institute for the Structure and Dynamics of Matter and Center for Free-Electron Laser Science, Hamburg, Germany. [2]Initiative for Computational Catalysis (ICC), The Flatiron Institute, New York, NY, USA. [3]Department of Physics, Technical University of Denmark, Kgs. Lyngby, Denmark. ✉e-mail: francesco.troisi@mpsd.mpg.de; hannes.huebener@mpsd.mpg.de; angel.rubio@mpsd.mpg.de; simola@dtu.dk

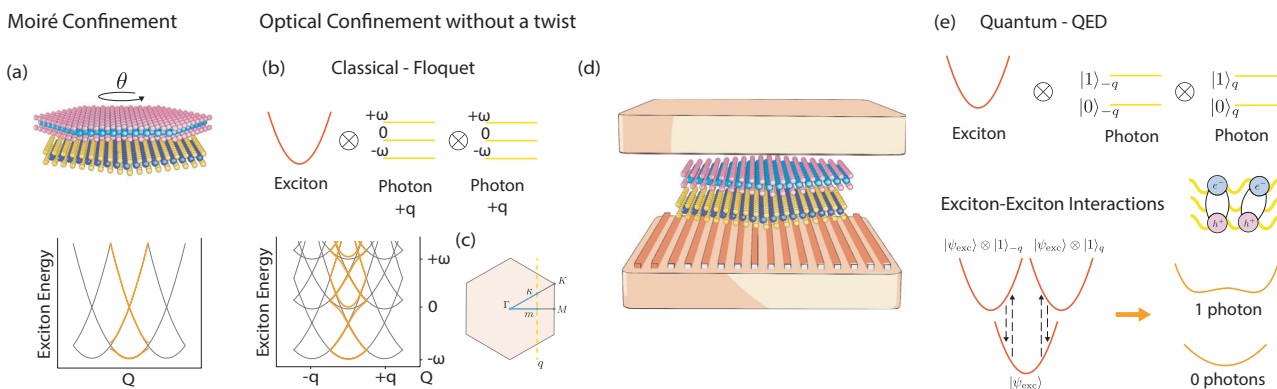

**Fig. 1 | Schematics of the Moiré band formation and cavity-mediated interaction in a TMD bilayer heterostructure. a** A structural twisted bilayer TMD heterostructure forms a Moiré super-lattice, leading to spatial modulation of the excitonic states. **b** Untwisted heterostructure embedded into a classically driven spatially structured cavity, the light-matter interaction generates an all-optical Moiré potential. **c** Representation of the $k$-points path used in the spectral function. We use the standard $M − \Gamma − K$ path, which we shorten to $m − \Gamma − \kappa$, where $m, \kappa$ are taken at the value of the photon momentum along the aforementioned path. **d** Representation of an untwisted TMD heterostructure embedded in a spatially structured optical cavity. **e** When the heterostructure is untwisted (no standard Moiré potential) and embedded into a dark spatially structured cavity (where the vacuum fluctuation plays a role), light-matter coupling mediates exciton-exciton interactions and renormalizes the excitonic mass.

We analyze the role of cavity-mediated interactions in a prototypical type-II MoSe$_2$/WSe$_2$ heterostructure embedded in a planar cavity setup. We consider both an unstructured planar cavity and a structured (grated) cavity. The former can be described as a single effective mode of the electromagnetic field, where light carries no momentum, while the latter requires a multi-mode description and allows for momentum transfer between light and matter. Our theoretical framework builds, on the one hand, on the methodology for the first-principles treatment of excitons in Moiré potentials established in ref. 14 and on the other hand, on a low-energy quantum electrodynamical (QED) Hamiltonian approach for the coupling of excitons to the cavity[18,35]. More specifically, we solve the Mott−Wannier equation in momentum space to obtain the low-energy excitonic states. Subsequently, we derive a low-energy QED Hamiltonian that can describe both the electrostatic Moiré potential arising from twisting the bilayer and the coupling to the photonic modes by using an excitonic representation of the many-body QED problem. Finally, we perform the full diagonalization of the QED Hamiltonian to obtain the hybrid exciton-polariton states, from which we predict the excitonic dispersion, via the excitonic spectral function, and the optical properties of the cavity-matter system by computing the interacting optical linear absorption spectra.

We find that spatially unstructured cavities, where the electromagnetic field carries no momentum, can alter the twist-induced excitonic confinement from the Moiré potential when the cavity mode is resonant with the excitonic transition. This result is in agreement with previous works[36], where they applied a Hopfield diagonalization to a tight-binding model to study how bare photonic and excitonic states contribute to exciton-polariton states in unstructured Fabry−Perot cavities at different twist angles. We show that the same result can be obtained using a more general theory, starting with a first-principles QED Hamiltonian, and focusing on controlling the localization of exciton-polaritons. Conversely, spatially structured cavities, where momentum can be exchanged between light and matter, can induce optical confinement and emulate the Moiré potential when driven with a classical electromagnetic field (i.e., a laser), even in the *absence* of twist in the embedded bilayer. For dark spatially structured cavities, instead, where the light-matter coupling arises from the quantum fluctuations of the electromagnetic field inside the cavity, we find the emergence of cavity-induced exciton-exciton interactions in untwisted bilayers. This leads to both excitonic confinement and mass renormalization and consequently to a modification of the optical properties of the material in equilibrium, going beyond what is possible to achieve with twisting.

## Results
### Theory
To study the behavior of Moiré excitons in an optical cavity, we first model the two uncoupled systems (excitons and photons) and subsequently describe their interaction. Then, we formulate and discuss the QED Hamiltonian for Moiré excitons, which constitutes one of the main results of the paper. All throughout this work, we use atomic units.

The uncoupled matter Hamiltonian $\hat{H}_M$ of a twisted bilayer heterostructure can be formulated in an excitonic representation following ref. 14 as

$$\hat{H}_M = \sum_{ll',\nu} \sum_{i\in C,j\in V} \sum_{\boldsymbol{Q}} \mathcal{E}^\nu_{ll',ij,\boldsymbol{Q}} \hat{X}^{\nu\dagger}_{ll',ij,\boldsymbol{Q}} \hat{X}^\nu_{ll',ij,\boldsymbol{Q}}$$
$$+ \sum_{ll',\nu} \sum_{i\in C,j\in V} \sum_{\boldsymbol{Q},\boldsymbol{q}} \mathcal{M}^\nu_{ll',ij,\boldsymbol{q}} \hat{X}^{\nu\dagger}_{ll',ij,\boldsymbol{Q}+\boldsymbol{q}} \hat{X}^\nu_{ll',ij,\boldsymbol{Q}} \quad (1)$$

where $i, j$ are band indexes which span over conduction ($C$) or valence ($V$) band states, and $l, l'$ are layer indexes. The operators $\hat{X}^\dagger, \hat{X}$ create (annihilate) an exciton between any pair of bands of either the same layer (when $l = l'$, in which case one has intralayer excitons) or different layers (interlayer excitons), $\boldsymbol{Q}$ is the momentum associated with the center of mass of the exciton, and $\boldsymbol{q}$ is the momentum transferred by the Moiré potential. The index $\nu$ refers to the bound state (i.e., $1s, 2s...$). $\mathcal{E}^\nu_{ll',ij,\boldsymbol{Q}} = \frac{\boldsymbol{Q}^2}{2m_{ll',ij}} + E_{g,ll'} + E^\nu_{b,ll'}$ encodes the dispersion relation of a free exciton, which we assume parabolic, where $m_{ll',ij}$ is the excitonic effective mass, $E_{g,ll'}$ is the energy gap and $E^\nu_{b,ll'}$ is the binding energy, which we obtain by solving the Mott−Wannier equation (c.f. Supplementary Information Section V.A). $\mathcal{M}$ is the matrix element of the Moiré potential in the excitonic basis, describing the Moiré scattering of excitons with different momenta. Note that $\mathcal{M}$ carries the information on the Mott−Wannier states (c.f. Supplementary Information Eq. A25). Refer to the Supplementary Information Section I for a detailed derivation.

We describe the uncoupled light system with a Hamiltonian consisting of a set of effective harmonic oscillators (the photon modes of

the cavity):

$$\hat{H}_{ph} = \sum_{\bar{q},\lambda} \omega_{\bar{q}} \left( \hat{a}^{\dagger}_{\bar{q},\lambda} \hat{a}_{\bar{q},\lambda} + \frac{1}{2} \right) \qquad (2)$$

where $\omega_{\bar{q}}$ represents the energy of the photon mode $\bar{q}$ and $\lambda$ is the polarization. $\bar{q}$ is the momentum of the photon in the $xy$ plane of the 2D material. In a planar setup, electromagnetic waves propagate in the in-plane direction, while they are standing waves in the $z$-direction (which is orthogonal to the cavity mirrors). We assume that the matter couples with the in-plane component of the electric field and therefore only consider modes for which it is finite. The momentum of the photon in the $z$-direction is set by the standing wave condition for the fundamental cavity mode. Additionally, when a periodic grating is present, the $xy$ plane momentum is finite and determined by the periodicity of the grating. When studying optical properties of materials in the far field, one usually makes the approximation that the light does not carry any momentum, the long-wavelength approximation (LWA). We recently showed that, also within a QED framework, the LWA is applicable[35] and reduces the description of the electromagnetic Hamiltonian to one with a single effective mode at $\bar{q} = 0$. In practice, the LWA implies that an electron cannot scatter to another $k$-point following the absorption of a photon (vertical transition in $k$ − space). In contrast, if the cavity is spatially structured with a grating, the photon modes can acquire a finite momentum $\bar{q}$ and couple matter excitations with different momenta. As shown later, this can give rise to a non-local interaction of excitons in $k$-space. Note that even though the construction of effective cavity modes is provided in ref. 35 is within the LWA, we here apply a similar procedure to construct the few effective collective modes in Eq. (2) for which the in-plane momenta are not averaged around $\bar{q} = 0$, but around the finite momenta $\bar{q}$ set by the cavity. We stress here that the term "effective", when referring to the cavity modes, indicates that such modes represent a summation over a continuum of modes centered around a certain momentum $\bar{q}$. Working with effective modes is necessary to guarantee a finite light-matter coupling strength in the bulk limit of extended cavity-matter systems when working with a finite number of modes for the description of the electromagnetic field.

To describe the light-matter coupling, we start by performing the canonical momentum substitution $\hat{p} \rightarrow \hat{p} + \hat{A}$ to the uncoupled matter Hamiltonian, obtaining the second-quantized Pauli-Fierz Hamiltonian[35]. The Hamiltonian in the excitonic representation reads (c.f. the Supplementary Information Section I for the complete derivation):

$$\hat{H} = \sum_{\bar{q},\lambda} \omega_{\bar{q}} \left( \hat{a}^{\dagger}_{\bar{q},\lambda} \hat{a}_{\bar{q},\lambda} + \frac{1}{2} \right) + \sum_{ll',\nu} \sum_{i \in C, j \in V} \sum_{\boldsymbol{Q}} \left( \mathcal{E}^{\nu}_{ll',ij,\boldsymbol{Q}} \hat{X}^{\nu\dagger}_{ll',ij,\boldsymbol{Q}} \hat{X}^{\nu}_{ll',ij,\boldsymbol{Q}} \right.$$
$$+ \sum_{\boldsymbol{q}} \mathcal{M}^{\nu}_{ll',ij,\boldsymbol{Q}\boldsymbol{q}} \hat{X}^{\nu\dagger}_{ll',ij,\boldsymbol{Q}+\boldsymbol{q}} \hat{X}^{\nu}_{ll',ij,\boldsymbol{Q}} + \sum_{\bar{q},\lambda} \tilde{A}_{0,\bar{q}} \sum_{ll',\nu} \sum_{i \in C, j \in V} \sum_{\boldsymbol{Q}} \qquad (3)$$
$$\left[ \mathcal{B}^{\nu\lambda}_{ll',ij,\boldsymbol{Q}\bar{q}} \hat{X}^{\nu\dagger}_{ll',ij,\boldsymbol{Q}+\bar{q}} \hat{X}^{\nu}_{ll',ij,\boldsymbol{Q}} \left( \hat{a}^{\dagger}_{\bar{q},\lambda} + \hat{a}_{-\bar{q},\lambda} \right) \right.$$
$$\left. + \mathcal{I}^{\nu,\lambda}_{ll',ij,\boldsymbol{Q}\bar{q}} \hat{X}^{\nu\dagger}_{ll',ij,\bar{q}} \left( \hat{a}^{\dagger}_{\bar{q},\lambda} + \hat{a}_{-\bar{q},\lambda} \right) + h.c. \right]$$

where $\tilde{A}_{0,\bar{q}} = \frac{A_{0,\bar{q}}}{\sqrt{V_{\text{eff},\bar{q}}}}$, $A_{0,\bar{q}}$ is the coupling strength of the mode $\bar{q}$ and $V_{\text{eff},\bar{q}}$ is the effective mode volume. The first line of the Hamiltonian contains the uncoupled photon and the uncoupled matter, while the second represents the paramagnetic bilinear coupling between photon modes and excitons. Note that we absorbed the diamagnetic term into the uncoupled photon Hamiltonian by performing a Bogoliubov transformation[37,38]. $\mathcal{B}$ and $\mathcal{I}$ are the matrix elements describing the coupling to the matter momenta[35] in momentum-conserving exciton-photon interactions and are defined in Supplementary Information Section I.C. The former allows an exciton to scatter to another $k$-point after absorbing or emitting a photon, the implications of which will be

discussed in depth in the next sections. Importantly, $\mathcal{B} = 0$ when $\bar{q} = \boldsymbol{0}$ (see Supplementary Information Section V.C). The matrix elements $\mathcal{I}$ couple the material ground state to the light by creating an exciton. Hence, while the term $\mathcal{B}$ explicitly conserves the number of excitons, $\mathcal{I}$ deals with the creation or destruction of such particles. In order to access the polaritonic states via full diagonalization, the Hamiltonian in Eq. (3) is projected onto a combined light-matter product state, with excitonic states and the many-body ground state for the matter[18], which are written in a Slater determinant representation, and number states for each photonic mode. This basis keeps the N-particles electronic states explicit, so that even if the number of excitons is not conserved, the total number of particles is fixed.

In Eq. (3), we observe that the standard Moiré potential $\mathcal{M}$ and the first term of the bilinear coupling $\mathcal{B}$ share the same excitonic operators $\hat{X}^{\dagger}, \hat{X}$, in case $\bar{q} = \boldsymbol{q}$. For the Moiré potential, $\boldsymbol{q}$ refers to the super-lattice Moiré periodicity. The bilinear coupling $\bar{q}$ refers to the periodicity of the electromagnetic field inside the cavity. Despite having different physical meanings, both indices determine a lattice periodicity, which modifies the symmetry experienced by the excitons. To highlight this equivalence, we let $\bar{q} = \boldsymbol{q}$ and rewrite the interaction terms of the Hamiltonian in Eq. (3) (which also includes the non-conserving bilinear coupling $\mathcal{I}$) as:

$$\hat{H}_{\text{int}} = \sum_{ll',\nu} \sum_{i \in C, j \in V} \sum_{\boldsymbol{Q},\boldsymbol{q}} \left[ \hat{X}^{\nu\dagger}_{ll',ij,\boldsymbol{Q}+\boldsymbol{q}} \hat{X}^{\nu}_{ll',ij,\boldsymbol{Q}} \right.$$
$$\left( \mathcal{M}^{\nu}_{ll',ij,\boldsymbol{Q},\boldsymbol{q}} + \tilde{A}_{0,\boldsymbol{q}} \sum_{\lambda} \mathcal{B}^{\nu\lambda}_{ll',ij,\boldsymbol{Q}\boldsymbol{q}} \left( \hat{a}^{\dagger}_{\boldsymbol{q},\lambda} + \hat{a}_{-\boldsymbol{q},\lambda} \right) \right) + \qquad (4)$$
$$\left. \tilde{A}_{0,\boldsymbol{q}} \sum_{\lambda} \mathcal{I}^{\nu,\lambda}_{ll',ij,\boldsymbol{Q}\boldsymbol{q}} \hat{X}^{\nu\dagger}_{ll',ij,\boldsymbol{q}} \left( \hat{a}^{\dagger}_{\boldsymbol{q},\lambda} + \hat{a}_{-\boldsymbol{q},\lambda} \right) + h.c. \right]$$

This similarity, and indeed mutual mathematical interchangeability, of the exciton-exciton interaction provided by the Moiré potential and the structured optical modes is the central theoretical result of this work. In the following section, we investigate the effect of $\hat{H}_{\text{int}}$ on the excitonic system when the radiation field is treated both in a classical and quantum manner.

In our simulations, we consider only the $1s$ MoSe$_2$ intralayer exciton in the hetero-bilayer and reduce the photon space to the Fock number states $\{|0\rangle, |1\rangle\}$. This choice is motivated by computational feasibility, but we expect the predicted phenomena to hold for interlayer excitons at even larger grating periodicities. Throughout this work, the cavity periodicity is always one-dimensional (see Fig. 1d). Finally, we note that for the following comparison, we either simulate the Moiré term $\mathcal{M}$ or the cavity periodicity $\mathcal{B}$, but we never consider both of them in the same simulation.

**Classically driven cavity**

We consider the effect of a driven structured cavity on the excitonic states of the untwisted bilayer material and show that it gives confinement signatures similar to a twist-induced Moiré potential, which include the formation of a Moiré Brillouin-Zone, and the opening of miniband gaps. Examining Eq. (4), we note that the bilinear coupling $\mathcal{B}$ shares the same excitonic operators of the Moiré potential $\mathcal{M}$, but it is additionally paired with photonic operators. In the case of classical radiation, these operators are replaced by their mean-field value. To describe the driven system, we assume that the far field driving can couple to finite momentum cavity modes ($\pm \bar{q}$) via the grating of the cavity. We model this excitation for single-polarization ($\lambda = s$) modes as a time-dependent coherent state $|\tilde{\lambda}_{\bar{q}} \tilde{\lambda}_{-\bar{q}}(t)\rangle = |\tilde{\lambda}_{\bar{q}}(t)\rangle \otimes |\tilde{\lambda}_{-\bar{q}}(t)\rangle$. Here we define the time-dependent coherent states as $|\tilde{\lambda}_{\bar{q}}(t)\rangle = e^{\frac{i\omega_{\bar{q}}t}{2}} |e^{-i\omega_{\bar{q}}t} \tilde{\lambda}_{\bar{q}}\rangle$ where the last ket goes by the usual definition $|\alpha\rangle = e^{-\frac{|\alpha|^2}{2}} \sum_{s=0}^{\infty} \frac{\alpha^s}{\sqrt{s!}} |s\rangle$ of coherent states. Using that $\omega_{\bar{q}} = \omega_{-\bar{q}}$ and $\tilde{A}_{0,\bar{q}} = \tilde{A}_{0,-\bar{q}}$, and projecting the interaction Hamiltonian in Eq. (4) onto these states introduces a

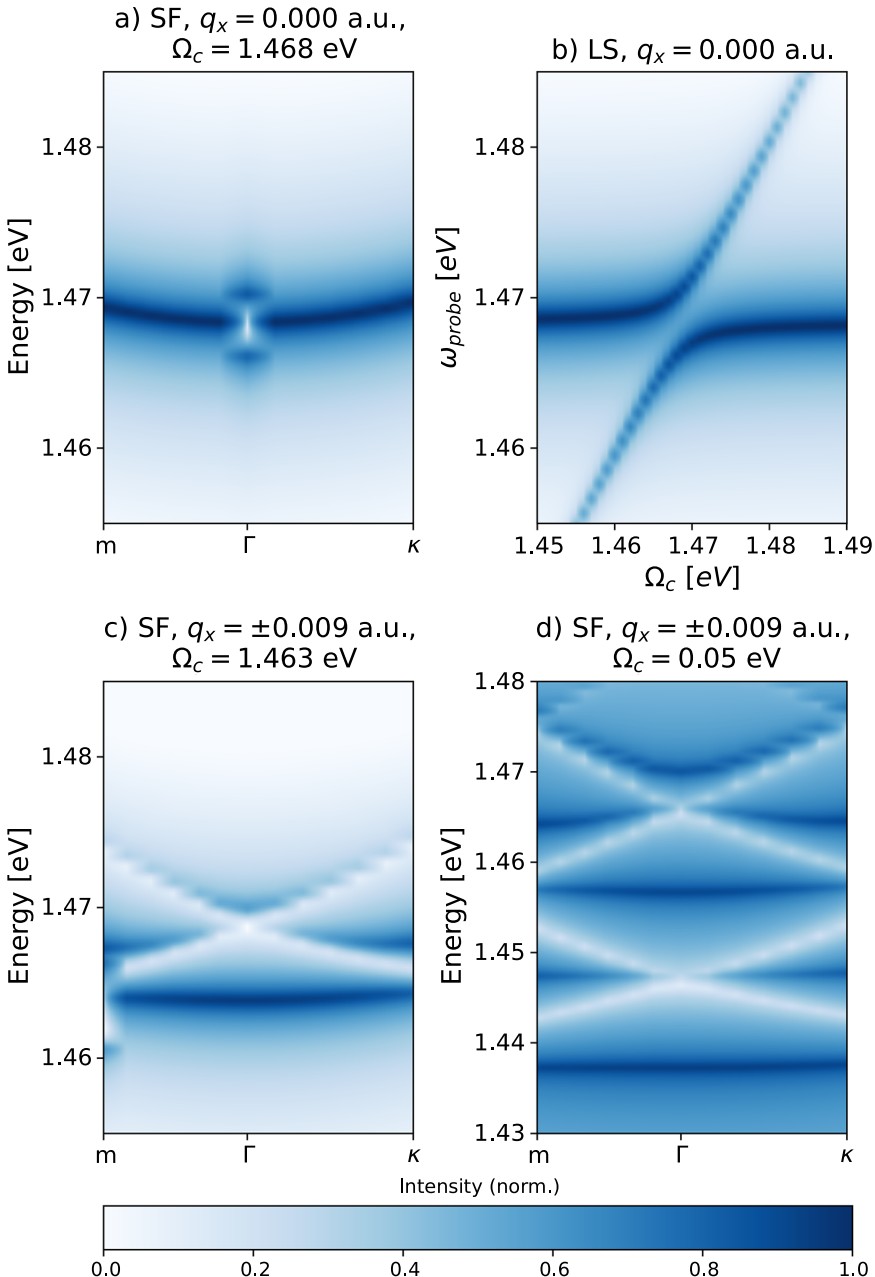

**Fig. 2 | Spectral function and imaginary part of the linear susceptibility for the untwisted heterostructure in a driven optical cavity (both structured and unstructured).** Spectral function (SF, c.f. Sec., **a**, **c**, **d**) and imaginary part of the linear susceptibility (LS, c.f. Sec., **b**) for the 1s intralayer exciton in the MoSe$_2$ layer when light can be treated classically and in the absence of twist-induced Moiré potential (in Eq. (4)). We use Floquet theory to solve the light-matter problem. We used $\tilde{A}_0 = 0.04$ a.u. for (**a**–**c**) and $\tilde{A}_0 = 0.02$ a.u. for (**d**). All panels use a normalized log scale for the intensity. For the spectral function, we use the standard $M - \Gamma - K$ path (which we shorten to $m - \Gamma - \kappa$, as shown in Fig. 1c and Supplementary Information Section II.B).

**a**, **b** The light does not carry any momentum, so no terms in the Hamiltonian connect two $k$-points. Hence, we observe a parabolic dispersion in the spectral function. In the linear response, we observe avoided crossing of the UP and LP formed from the excitonic state. **c**, **d** We used two photon modes carrying momentum $q_x = \pm 0.009$ a.u. In this case, the bilinear coupling generates a confining potential, and we observe the folding of the bands. (**c**) is at excitonic resonance (we can observe the Rabi splitting). The value slightly differs from (**a**) due to the bilinear coupling $\mathcal{B}$. On the contrary, (**d**) is at off-resonance, but the spectrum is rich due to the mixing of the Floquet replicas. The value of $\Omega_c = 0.05$ eV ensures the mixing of Floquet replicas.

time dependence of the form:

$$\left\langle \tilde{\lambda}_{\bar{q}} \tilde{\lambda}_{-\bar{q}}(t) | \hat{H}_{\text{int}} | \tilde{\lambda}_{\bar{q}} \tilde{\lambda}_{-\bar{q}}(t) \right\rangle \sim H^{(n=1)} e^{i\omega_{\bar{q}} t} + H^{(n=-1)} e^{-i\omega_{\bar{q}} t}$$

where $H^{(n=\pm 1)}$ is the time-*independent* projection of the interaction Hamiltonian onto the coherent state, with $n$ the Floquet frequency index. See Supplementary Information Section III for the complete expression.

Since the obtained Hamiltonian is time periodic, we use Floquet theory to solve it[39,40]. The advantage of this approach is that the Floquet Hamiltonian is time independent, which makes it computationally cheaper while still being able to capture the polaritonic effects from the light-matter coupling. With this semi-classical treatment, light with finite momentum $\bar{q}$ enters the coupled Hamiltonian in Eq. (4) exactly as the potential created by a Moiré super-lattice, by generating interacting Floquet replicas, as sketched in Fig. 1b. This means that we

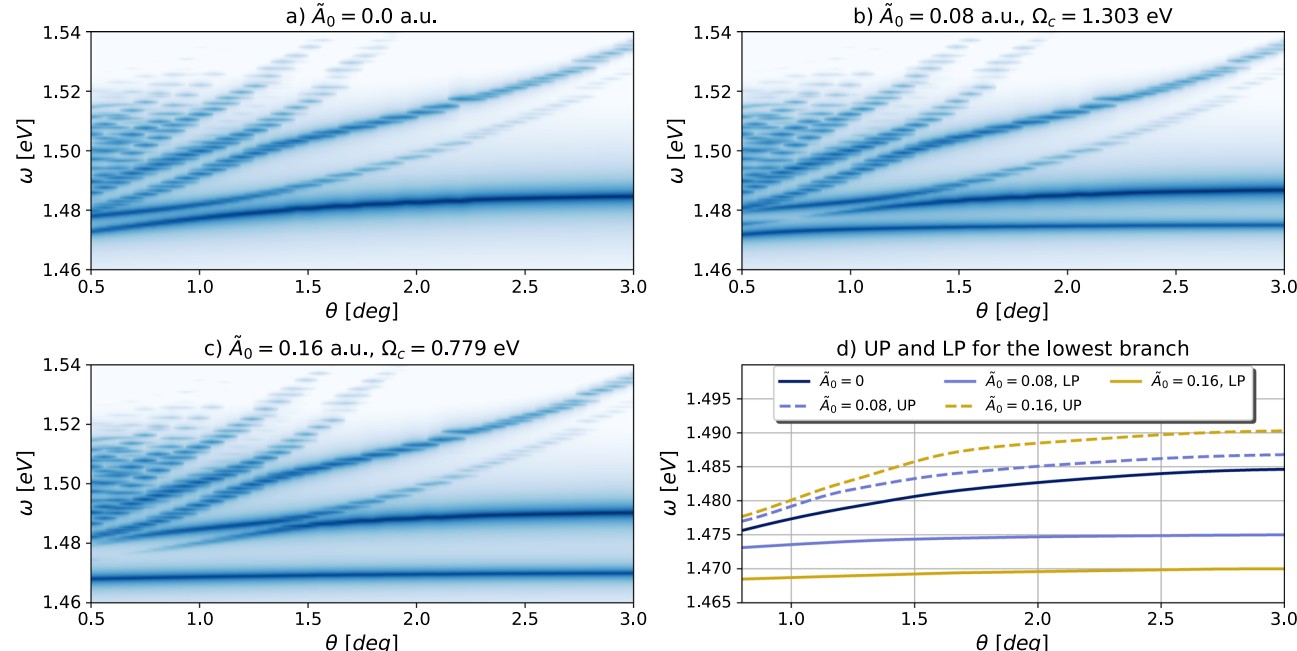

**Fig. 3 | Imaginary part of the linear susceptibility for the 1s intralayer exciton in the MoSe$_2$ layer, as a function of the twist angle.** The spacing used for the grid allows for a resolution of at most $\Delta\theta = 0.05$ deg, which explains the coarse nature of the lines (especially at small angles). (**a**–**c**) use a normalized log scale for the intensity (see Fig. 2). **a** Twisted heterostructure without the cavity (see ref. 14 Fig. 5a and ref. 36 Fig. 4a). **b, c** Twisted heterostructure embedded in a spatially unstructured cavity ($\bar{q} = 0$) for different light-matter coupling strengths. After fixing the energy of the mode to a constant value for all twist angles (in resonance with the lowest branch), we scanned over $\theta$. While all other states are mostly unaffected by the cavity, the lowest branch is split into the Upper (UP) and Lower (LP)

Polariton, and the separation increases with the light-matter coupling $\tilde{A}_0$. Note that for these two panels, $\Omega_c$ is significantly smaller than the excitonic resonance due to the diamagnetic shift. These results are qualitatively in agreement with ref. 36 Fig. 4b. **d** Trace of the UP and LP formed from the lowest branch of the bare excitonic system (taken from the previous panels). The separation between UP and LP increases with the light-matter coupling. Furthermore, the LP is almost flat for all twist angles, meaning it is mainly unperturbed by the Moiré potential in the presence of a cavity, whereas the UP is affected by the Moiré potential only at low angles.

can use classically driven cavities to induce an all-optical Moiré potential.

Figure 2 shows the results for a classically driven unstructured cavity as well as for a spatially structured cavity (with a one-dimensional spatial periodicity), both coupled to the 1s intralayer exciton in the MoSe$_2$ layer within the untwisted heterostructure (i.e., $\mathcal{M} = 0$). For the unstructured cavity (Fig. 2a, b), where we study the system at the resonance between the cavity mode and the exciton, the dispersion relation of the exciton remains parabolic, as seen in the spectral function (SF, c.f. "Methods" section for its definition) in Fig. 2a. On top of the unperturbed dispersion, we observe a signature of the formation of an exciton-polariton by the appearance of a splitting at Γ, corroborated by the expected avoided crossing of upper and lower polariton branches in the imaginary part of the linear susceptibility (LS, c.f. "Methods" section for its definition) as a function of cavity energies, our proxy for an absorption experiment (c.f. Fig. 2b). We then shift focus to a structured cavity both at and out of resonance with the excitons (Fig. 2c, d). Here, the excitons experience a confining potential generated by the bilinear coupling, which modifies the excitonic bands. Specifically, at resonance (Fig. 2c), we observe a simple band folding stemming from the periodicity of the grating together with the mixing with the finite momentum replica of the many-body ground state. The polariton splitting is now shifted to finite momentum (the *m*-point), as compared to Fig. 2a. Off-resonance, at small frequencies (Fig. 2d), the positive and negative excitonic replicas mix, generating a rich spectrum. We find that to achieve this mixing the cavity energy should be such that the excitonic replicas can intersect within the BZ. Finally, we find that at large off-resonant driving frequencies (not shown), the external driving does not induce any modification to the parabolic dispersion. This is because within the first-order high-frequency expansion of the Floquet

Hamiltonian[39], $H^{\omega\to\infty}_{\text{Floquet}} = H^{(n=0)} + \frac{[H^{(n=-1)}, H^{(n=1)}]}{\Omega} + \mathcal{O}(\frac{1}{\Omega^2})$, the commutator in the numerator vanishes, i.e., $[H^{(n=-1)}, H^{(n=1)}] = 0$. In the Supplementary Information Section III.A, we show the explicit calculation of the commutator as well as the spectral function for various values of the cavity frequency.

## Dark cavity

We now analyze the cavity-exciton system for a dark cavity, i.e., in the absence of external fields, where the effect of the photons on the matter is via their quantum fluctuations. We begin by focusing on the quantum effects on the heterostructure embedded in a spatially unstructured cavity ($\mathcal{B} = 0$), including in this case the periodic confining potential from the Moiré pattern caused by twisting. In the absence of momentum exchange, light and matter are coupled only through the bilinear coupling associated with $\mathcal{I}$ in Eq. (4). Here, we examine how the unstructured planar cavity influences the twist-angle dependence of the excitonic response. For reference, in Fig. 3a–c, we show the imaginary part of the linear susceptibility (c.f. "Methods" section for its definition) as a function of the twist angle $\theta$ and the optical probe energy $\omega$ outside the cavity. As the twist angle decreases, the Moiré Brillouin-zone contracts, the excitonic bands fold closer to the Γ point, and more folded states enter the investigated energy window. Therefore, the resulting linear response displays a richer spectrum at small twist angles, in line with earlier findings[14,36].

In the presence of cavity-matter coupling, when the cavity resonates with the MoSe$_2$ excitonic transition, we expect the Rabi splitting to dominate the behavior of the exciton-polariton response. We observe that compared to the case of uncoupled light and matter (Fig. 3a), the coupling splits the lowest branch into two, the upper (UP) and lower polariton (LP) (Fig. 3b, c). The magnitude of this splitting

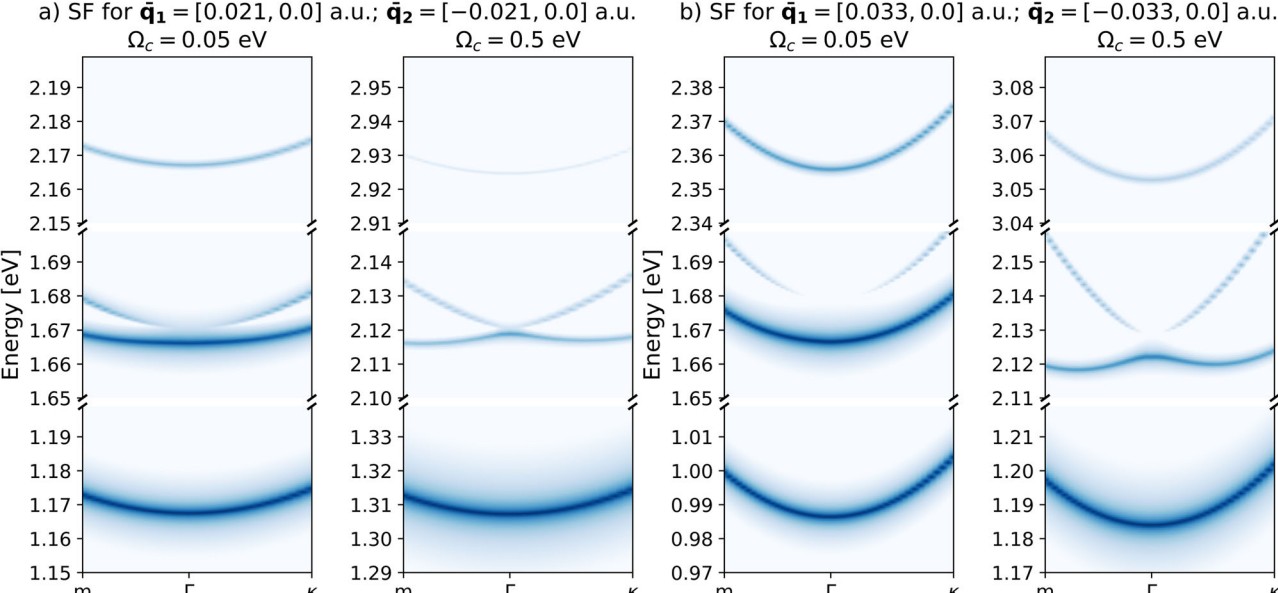

**Fig. 4 | Spectral function for the untwisted heterostructure ($\mathcal{M} = 0$ in Eq. (4)) in a dark spatially structured optical cavity with two photonic modes.** On the x-axis, we report the k-point. m and κ are the borders of the BZ with periodicity induced by the momentum carried by light (see Supplementary Information Section II.B). We used $\tilde{A}_{0,\bar{q}} = 0.08$ a.u. for all panels. All panels use a normalized log scale for the intensity (see Fig. 2). Note that the separation between the three bands depends both on the cavity energy $\Omega_c$, on the diamagnetic term $\tilde{A}_{0,\bar{q}}^2$, and on the interaction energy $\tilde{A}_{0,\bar{q}} \mathcal{B}_{ll',ij,Q\bar{q}}^{\nu,\lambda}$. Finally, the position of the bottommost band is shifted towards lower energies due to the interaction energy contribution.

**a** Spectral function for $\bar{q}_{1,2} = [\pm 0.021, 0.0]$ a.u., for both low and high frequency. In

both cases, the curvature of the dispersion relation in the uppermost and lowermost bands is small, a signature of a heavy exciton. In the central band instead, the dispersion relation goes from parabolic to M-shaped, which means that the excitonic mass is negative around the Γ point. **b** Spectral function for $\bar{q}_{1,2} = [\pm 0.033, 0.0]$ a.u., for both low and high frequency. In both cases, the curvature of the dispersion relation in the uppermost and lowermost bands is steep, a signature of a light exciton. Similarly to (**a**), the dispersion relation in the central band goes from parabolic to M-shaped, which means that the excitonic mass is negative around the Γ point.

increases linearly with the light-matter coupling $\tilde{A}_0$. Interestingly, the UP in Fig. 3b, c has the same dispersion as in the uncoupled matter system (Fig. 3a), whereas the LP line is almost constant for all twist angles, in agreement with previous works[36]. This is shown in Fig. 3d, where we traced out the curves obtained by the UP and the LP of Fig. 3b, c together with the lowest branch of the bare excitonic system (Fig. 3a). A flat line in the spectrum means that the twist angle, which controls the size of the Moire Brillouin-Zone, has little influence on the excitonic states. This implies that the LP behaves as a low-energy unperturbed and unconfined exciton. Conversely, the dispersion of the UP is steeper and linear at small twist angles and then flattens out for larger angles, where it behaves similarly to the LP, i.e., as nearly free particles. Ref. 36 provides a complementary discussion on how bare photonic and excitonic states contribute to the UP and LP. In short, the effect of the twisting angle can be modified by the electromagnetic field of a cavity, which influences the excitonic response by causing Rabi splitting of the resonant transition. This results in a different localization for the upper and lower polariton, and may pave the way for new quantum phases[41].

We then consider a spatially structured cavity, where both creation of excitons and Moiré-like exciton-scattering are included, i.e., both $\mathcal{B}$ and $\mathcal{I}$ are finite in Eq. (3), while we let $\mathcal{M} = 0$ to isolate the effect of the periodic grating of the cavity. First, we analyze the results in Fig. 4, which shows the excitonic spectral function. In all panels, there are three different sets of bands. From the analysis of the polaritonic states, resulting from the diagonalization of Eq. (3), we identify the lowest energy sector to be mainly composed of an uncoupled excitonic state and the vacuum ($n = 0$) state of both photonic modes. The central one is mainly constituted by an uncoupled excitonic state and one photon excitation ($n = 1$) in either of the modes. Finally, the high-energy band corresponds to an uncoupled excitonic state and a single excitation in both photon modes. The separation between the three bands increases with the cavity energy $\Omega_c$, with the diamagnetic term $\tilde{A}_{0,\bar{q}}^2$ and with the

interaction energy $\tilde{A}_{0,\bar{q}} \mathcal{B}_{ll',ij,Q\bar{q}}^{\nu,\lambda}$. Inspecting the uppermost and bottommost energy bands of Fig. 4, the curvature of the dispersion relation increases with the transfer of cavity photon momentum. As the spectral function is related to the excitonic band structure, we can interpret this variation as the modification of the excitonic mass. Since a steeper curvature corresponds to a smaller mass, the exciton becomes lighter as the photon momentum increases. In the central band instead, the dispersion relation goes from parabolic to M-shaped (i.e., the Γ point is a local maximum instead of a global minimum). This implies that the excitonic mass is negative around this point, and might lead to the decay of the zero-momentum exciton towards one of the local minima, creating a new stable exciton-polariton state at finite momentum. It should be noted that for the central band to show such features, the condition $\tilde{A}_{0,\bar{q}} \mathcal{B}_{ll',ij,Q\bar{q}}^{\nu,\lambda} \approx \tilde{A}_{0,-\bar{q}} \mathcal{B}_{ll',ij,Q,-\bar{q}}^{\nu,\lambda}$ should hold (i.e., both modes have the same interaction energy). Otherwise, the two exciton-polaritons would exist at different energy scales and could not scatter. To prove that this effect could be experimentally demonstrated, we show in Fig. 5 the linear susceptibility as a function of probe photon momentum. In this figure, going from left to right corresponds to going from Γ to m in Fig. 4b. As one can see, this observable is also able to capture the features discussed for the spectral function.

To better understand the physics, let us compare the effect of the bilinear coupling with that of the Moiré potential. In general, they both allow an exciton to scatter between different k-points. However, the different origin of the two terms plays a fundamental role in the effect on the excitonic system. The Moiré potential $\mathcal{M}$ acts as a periodic scattering potential, which allows momentum transfer between excitons. Conversely, the *optical* Moiré term $\mathcal{B}$, while also allowing the hopping of an exciton between k-points, does so at the expense of creating/annihilating a cavity photon. The underlying physics becomes even clearer when the excitons and the photons are off-resonant, so that photons can only be created/destroyed virtually. In this situation,

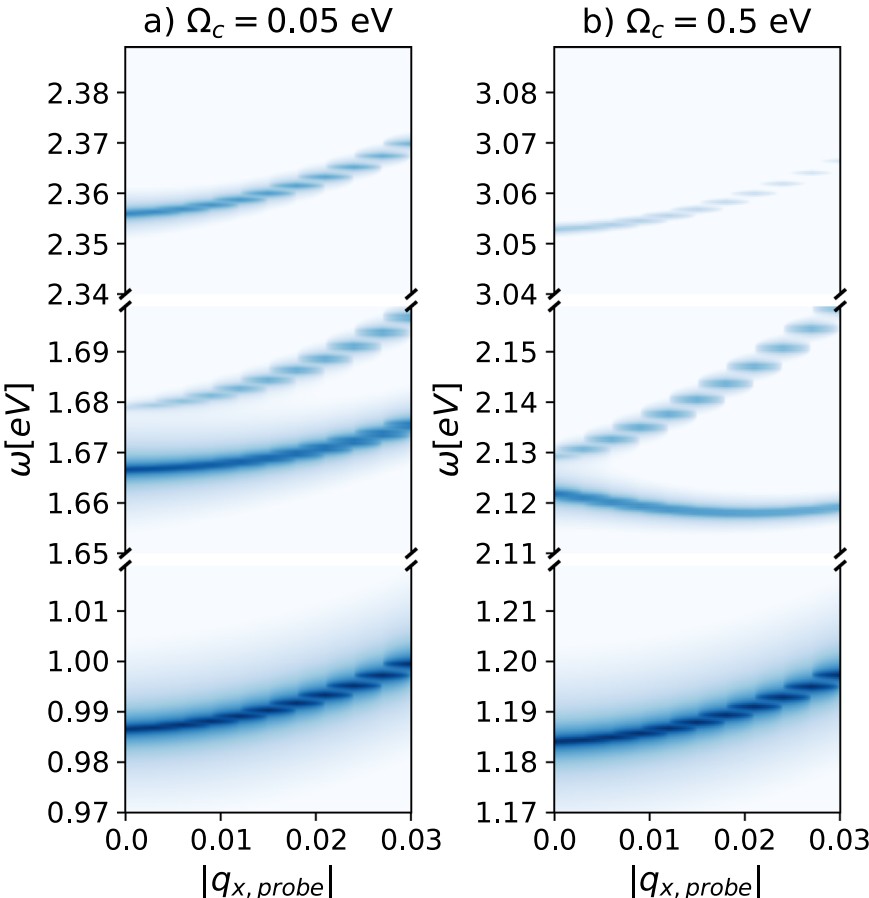

**Fig. 5 | Imaginary part of the linear susceptibility at finite $q$ for the untwisted heterostructure ($\mathcal{M} = 0$) in a dark spatially structured optical cavity with two photonic modes with $\bar{q}_1 = [0.033, 0.0]$ a.u.; $\bar{q}_2 = [−0.033, 0.0]$ a.u.** We used $\bar{A}_{0,q} = 0.08$ a.u. for all panels. All panels use a normalized log scale for the intensity (see Fig. 2). On the $x$-axis, we report the absolute value of the momentum carried by the probe beam, while on the $y$-axis, its energy. $|q_{x,\text{probe}}| = 0.01$ a.u. means that the probe is composed of two modes, one with $q_1 = [0.01, 0.0]$ a.u.; $q_2 = [−0.01, 0.0]$ a.u. This figure should be compared with Fig. 4b. While the first and third bands maintain the parabolic dispersion regardless of the energy of the cavity, the central one visibly changes. **a** Low-frequency case. Here, the central band is parabolic. **b** High-frequency case. Here, the central band has a maximum at $|q_{x,\text{probe}}| = 0.0$ a.u., which can be interpreted as a negative excitonic mass (similarly to Fig. 4).

the *optical* Moiré term turns into a many-body, exciton-exciton interaction, whose momentum dependence is solely determined by the cavity design. To explicitly show this feature, we make use of the high-frequency limit of the QED Hamiltonian. Within this limit, we can write a photon-free QED Hamiltonian by downfolding the original one in a dressed excitonic space according to refs. 30,38,42,43:

$$\hat{H}_{\text{QED}}^{\omega \to \infty} \approx \left\langle 0_{\bar{q}}, 0_{-\bar{q}} | \hat{H} | 0_{\bar{q}}, 0_{-\bar{q}} \right\rangle$$
$$-\sum_{\bar{q}} \frac{1}{\omega_{\bar{q}}} \left[ \left\langle 0_{\bar{q}}, 0_{-\bar{q}} | \hat{H} | 1_{\bar{q}}, 0_{-\bar{q}} \right\rangle \cdot \left\langle 1_{\bar{q}}, 0_{-\bar{q}} | \hat{H} | 0_{\bar{q}}, 0_{-\bar{q}} \right\rangle \right]. \tag{5}$$

Performing this expansion on the interaction Hamiltonian in Eq. (4) leads to (see Supplementary Information Section IV for the full derivation):

$$\hat{H}_{\text{int}}^{\omega \to \infty} = \sum_{ll',\nu} \sum_{i \in C, j \in V} \sum_{Q} \mathcal{M}_{ll',ij,Qq}^{\nu,\lambda} \hat{X}_{ll',ij,Q+q}^{\nu\dagger} \hat{X}_{ll',ij,Q}^{\nu}$$
$$-\frac{2\bar{A}_{0,q}\bar{A}_{0,-q}}{\omega_q} \sum_{ll_1l'} \sum_{iji_1j_1} \sum_{QQ_1,\nu\nu_1} \left( \mathcal{B}_{ll',ij,Q,-q}^{\nu,\lambda} \mathcal{B}_{l_1l',i_1j_1,Q_1q}^{\nu_1,\lambda} \right. \tag{6}$$
$$\hat{X}_{ll',ij,Q-q}^{\nu\dagger} \hat{X}_{l_1l',i_1j_1,Q_1+q}^{\nu_1} \hat{X}_{ll',ij,Q}^{\nu} \hat{X}_{l_1l',i_1j_1,Q_1}^{\nu_1}$$
$$\left. -\sum_q \mathcal{I}_{ll',ij,Qq}^{\nu,\lambda} \mathcal{T}_{l_1l',i_1j_1,Q_1q}^{\nu_1,\lambda,*} \hat{X}_{ll',ij,Q}^{\nu\dagger} \hat{X}_{l_1l',i_1j_1,Q_1}^{\nu_1} \right)$$

The striking result is the emergence of a four-excitonic operator term, i.e., a term that describes an interaction between two excitons, which is a fundamental difference with respect to the Moiré potential. This alters the excitonic dynamics as manifested by our calculated spectral functions and the non-trivial linear response features. These excitons-excitons interactions, absent in classical treatments, highlight the potential of quantum cavities to engineer correlated excitonic phases.

## Discussion

In this work, we demonstrated that spatially structured optical cavities can precisely emulate Moiré-like excitonic confinement in Van der Waals heterostructures, circumventing the need for physical lattice twisting.

For classically light-driven cavities, momentum-carrying modes replicate the spatial modulation of a standard Moiré potential, leading to band folding and splitting, typical of a free particle in a periodic potential. Such effects are predicted to be in the meV energy scale (for the parameters we used).

Conversely, in dark cavities, the light-matter interaction is mediated solely by quantum vacuum fluctuations of the cavity modes, which induce long-range exciton-exciton interactions, fundamentally altering the excitonic dynamics. Note that we have recently observed that in Graphene cavity can induce long-range interaction for electrons[38]. These interactions, not present when the system interacts with classical light, highlight the potential of quantum cavities to engineer correlated excitonic phases. Most notably, we obtain a negative excitonic mass around the Γ point that could be an

experimental fingerprint of the cavity-mediated excitonic interactions predicted in this work.

Our findings bridge the gap between twist engineering of two-dimensional heterostructures and cavity quantum electrodynamics, providing a versatile platform for optically programmable excitonic systems.

## Methods

In this work, we distinguish between spatially unstructured cavities, of the Fabry–Perot kind, where the electromagnetic field is treated with a single effective mode description in the long-wavelength approximation[35] ($\bar{q} = [q_x, q_y] = [0, 0]$), and spatially structured cavities, whose description requires two effective momentum-carrying modes ($\bar{q}_1 = [q_x, 0]$, $\bar{q}_2 = [-q_x, 0]$) corresponding to a cavity with a one-dimensional periodicity (like the one in Fig. 1d). Based on symmetry arguments, we assume that $\omega_{\bar{q}_1} = \omega_{\bar{q}_2}$ and $\tilde{A}_{0, \bar{q}_1} = \tilde{A}_{0, \bar{q}_2}$. Such photonic environments can be realized by means of dielectric metasurfaces[44,45], or in polaritonic and plasmonic cavities[46–48]. In order to illustrate the broad potential of cavity-structuring of excitons, we leave the cavity mode energy, volume, and grating wavelength as adjustable parameters.

In order to obtain the values of the modes' momentum used in the previous sections ($\bar{q}_i = [\pm 0.009, 0.0], [\pm 0.021, 0.0], [\pm 0.033, 0.0]$ a.u.), one should design the grating of the cavity accordingly. We estimate that the periodicity of the corresponding grating should be $d = \frac{2\pi}{\bar{q}_i} = 40$ nm, 16 nm, 10 nm. We expect these extreme grating sizes to be larger for the optical confinement of interlayer excitons.

In order to obtain the full exciton-polariton states, we start by solving the Mott–Wannier equation in the $k$-space (c.f. Supplementary Information Section V.A), which gives us access to the binding energy and the Mott–Wannier states. We considered only the $1s$ intralayer exciton in the MoSe$_2$ layer. The Mott–Wannier states are used to construct the excitonic creation and annihilation operators, as well as all the matrix elements appearing in Eq. (3). Refer to Supplementary Information Section I for the complete derivation. We represent the photonic modes using the Fock number states $\{|0\rangle, |1\rangle\}$. Finally, we obtain the full exciton-polariton states via the full diagonalization of Eq. (3). The physics of the coupled light-matter system is then investigated by computing excitonic quantities: the linear optical susceptibility (LS) function and the spectral function (SF). The former represents the optical response of the system, obtained from applying linear response theory to the polaritonic states[49]:

$$\chi(\omega, \Omega_c, \theta) = \sum_I \frac{|\mathcal{M}_{I,0}|^2}{\omega - (E_I(\Omega_c, \theta) - E_0(\Omega_c, \theta)) + i\eta} \tag{7}$$

where $\mathcal{M}_{I,0}$ is the transition matrix element between the polaritonic ground state $\Psi_0$ and an excited state $\Psi_I$. $\eta$ is a small artificial broadening, $\Omega_c$ is the energy of the cavity modes, $\omega$ is the energy of the probe field, $\theta$ is the Moiré twist angle, and $E_I(\Omega_c, \theta)$ is the energy of the $I$th polaritonic state. Note that when we set $\mathcal{M} = 0$ (no standard Moiré potential), $\chi$ does not depend on $\theta$. Furthermore, note that we only calculate the matter part of such a response by tracing out the photons. To investigate the excitonic dispersion, i.e., their band structure, we employ the spectral function defined as:

$$\mathcal{S}(\omega, \Omega_c, \theta, \boldsymbol{Q}) = \sum_I \frac{\langle \Psi_I | \hat{X}_{\boldsymbol{Q}}^\dagger | \Psi_0 \rangle \langle \Psi_0 | \hat{X}_{\boldsymbol{Q}} | \Psi_I \rangle}{\omega - (E_I(\Omega_c, \theta) - E_0(\Omega_c, \theta)) + i\eta} \tag{8}$$

This represents the probability of creating an exciton with an energy $E_I(\Omega_c, \theta)$ at a certain $k$-point. Note that when we set $\mathcal{M} = 0$ (no standard Moiré potential), $\mathcal{S}$ does not depend on $\theta$. Refer to the Supplementary Information Section II for additional insights.

## Data availability

The data that support the findings of this study are available from the corresponding authors upon request.

## Code availability

The code that supports the findings of this study is available from the corresponding authors upon request.

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

## Acknowledgements

We acknowledge support from the Villum foundation grant No. 72146 (S.L.), the Cluster of Excellence "CUI: Advanced Imaging of Matter" - EXC 2056 - project ID 390715994 (A.R.), European Research Council (ERC-2024-SyG-101167294; UnMySt) (A.R.) and Grupos Consolidados (IT1453-22) (A.R.), and the Max Planck-New York City Center for Non-Equilibrium Quantum Phenomena. We acknowledge support from the European Union Marie Sklodowska-Curie Doctoral Networks TIMES grant No. 101118915 (A.R.) and SPARKLE grant No. 101169225 (S.L.). The Flatiron Institute is a division of the Simons Foundation.

## Author contributions

F.T. performed all calculations and implementations, analyzed the data, and wrote the first draft of the manuscript. H.H. and A.R. contributed to the discussion and interpretation of the data. S.L. contributed to the original idea of the paper, as well as to the discussion and interpretation of the data. All authors contributed to the final version of the manuscript.

## Funding

## Competing interests

The authors declare no competing interests.
