## [Transparent Peer Review file · Nature Communications]

Cavity-QED-controlled two-dimensional Moiré Excitons without twisting

Corresponding Author: Mr Francesco Troisi

Version 0:

Reviewer comments:

Reviewer #1

(Remarks to the Author)

The authors propose an all-optical realization of Moiré-like exciton confinement using spatially periodic optical cavities. The idea of engineering Moiré physics in a purely photonic way is very appealing, as it avoids complications from atomic reconstruction and offers a higher degree of tunability. I also find the analysis of the dark cavity scenario particularly interesting and original.

Overall, I believe this work will attract broad interest at the interface of the photonic and Moiré physics communities. I therefore recommend publication in Nature Communications after the authors address the following points:

i) The authors correctly point out that the bilinear coupling term formally shares the same structure as the Moiré Hamiltonian, leading to an additional periodicity. However, for Moiré physics of real interest to emerge, the induced potential must be strong enough to produce flat bands, i.e., to substantially modify the band curvature, with miniband separations large compared to the exciton thermal distribution. Otherwise, excitons will experience a spatial modulation without true localization.

In the present case, the folding vector q is small, so the ground state appears flat only because it is very close to the parabolic minimum, and already the third band in Fig. 2c shows a more pronounced dispersion. This raises the question: to what extent does localization actually occur, given that at low temperatures a thermal distribution spans ~ 20 meV, whereas the miniband separation in Fig. 2c is less than 10 meV?

ii) Related to the above, if one increases the cavity parameters to enhance the strength of the bilinear term to overcome the point mentioned above, is it still valid to use the unmodified Wannier exciton states to define the exciton basis? In Moiré systems, the effective potential originates from atomic displacements and does not significantly alter the underlying Wannier exciton problem in first approximation, making it possible to treat it via an effective static continuum potential. By contrast, in the photonic case the periodic potential arises from light–matter interaction itself, and stronger coupling may renormalize the exciton wavefunctions directly. Strictly speaking, such a term should then be included in the Wannier equation to obtain new exciton states. I do not expect the authors to perform such a calculation, but I suggest that they comment on the range of validity of their approximation.

iii) At the end of the first paragraph of the “Dark cavity” section, the authors cite only Ref. 14. I suggest also citing Nano Lett. 2022, 22, 11, 4468–4474, where exciton–polariton Moiré structures were computed. Although that work addresses a non-dark cavity, it provides a useful comparison and would strengthen the discussion.

The authors might find the following list of minor corrections useful (some may be due to PDF conversion):

Page 1, first column, first line: "Van der Waals" → "van der Waals".

Page 5, second column, last line: "form" → "from".

Use of hyphens: "hetero-structure" → "heterostructure", "bi-layer" → "bilayer", "bi-linear" → "bilinear", "intra-layer" → "intralayer".

In the SI, Eqs. (17) and (19): the exciton annihilation operator is missing the index v .

Reviewer #2

(Remarks to the Author)

This work proposes and theoretically demonstrates an original approach for creating moiré systems without twists using structured planar optical cavities. The proposed schemes uses a TMD heterostructure embedded in a planar cavity with faces parallel to the structure. The authors show that, in the case of a structured cavity, the excitonic dispersion relations have characteristics of a moiré. The authors study the case of cavities driven by a classical field or without driving. In the second case, the interaction between the van der Waals heterostructure and the cavity induces long-range interactions.

This is an original approach, likely to open a new avenue in the field of moiré systems, avoiding twisting of solid-state structures. Moiré systems are attracting growing attention and the field is very active at the interface of condensed matter physics, quantum optics, and quantum gases. This new work enriches it and I am convinced that it will be of great interest to the community. The presentation of the theoretical model is clear and sound. Useful technical details of the calculations are well described in the supplementary material. In addition, the paper is clearly written and will be accessible even for non-specialists in the field.

I recommend the publication of the manuscript after the authors have considered the following points:

In section II.A, it would be useful to say something about the method used in numerical calculations.

It is written that the results "show that it gives confinement signatures similar to a twist induced Moiré potential" but the authors do not write precisely what makes the signature unambiguous. A discussion of these signatures would be very useful, particularly for the non-specialist reader.

It is also written that "B shares the same excitonic operators of the Moiré potential M" but in a paragraph where it is written that $M=0$ a few lines above. This is very misleading.

A few lines below eq 2, please specify that z is the direction orthogonal to the cavity planes.

The authors write "We assume that the matter couples with the in-plane component of the electric field and therefore only consider modes for which it is finite." What does justify this assumption? To what extent is this justified in practical implementation?

Page 6, right-hand-side column, it is written "In the range of small twist angles, however, we can identify localized polaritonic states [14]." What does induce localization? In fact, I am surprised that the localization appears at low angles but apparently not at larger angles, according to the text. I would expect the following phenomenology: For certain particular angles, the system is periodic and cannot induce localization; For most angles, on the other hand, the system is non-periodic and can induce localization. For very low angles but inducing a periodic structure, the moiré period is very large so that the couplings between subsequent cells are very weak and very small fluctuations can overcome them and give a false apparent localization. This was recently analyzed in a slightly different context in PRR 6, L042066 (2024) and PRA 111, 043305 (2025). Can the results observed in the manuscript be explained in a similar way?

Version 1:

Reviewer comments:

Reviewer #1

(Remarks to the Author)

The authors have thoroughly addressed all the comments raised in my previous review. The clarifications and revisions improve the presentation of the manuscript and provide sufficient detail regarding the methodology. I find the current version satisfactory and have no further comments. I therefore recommend the manuscript for publication in its present form.

Reviewer #2

(Remarks to the Author)

The authors have carefully answered all my initial concerns and clarified all points. This is a very interesting and useful work. I fully support publication in the present form.

Responses to the reviewers comments

We sincerely thank the two reviewers for their efforts to carefully evaluate our manuscript. We are very pleased that both reviewers recognize the importance, significance and validity of our work, and support the conditional acceptance of the paper. Their constructive and insightful comments are helpful for us to improve the quality of this work and clarify some points in the manuscript. Below, we provide our responses, in blue text, to each of the individual comments, along with a list of changes we made in the revised manuscript. We also thank the Editor for their consideration of our work and for inviting us for a revision. We added the *Data availability* and *Code availability* sections as requested by the Editor. Additionally, we added the *Authors contributions* and the *Competing Interests* sections to comply with the editorial policies.

1 Reply to Reviewer 1:

The authors propose an all-optical realization of Moiré-like exciton confinement using spatially periodic optical cavities. The idea of engineering Moiré physics in a purely photonic way is very appealing, as it avoids complications from atomic reconstruction and offers a higher degree of tunability. I also find the analysis of the dark cavity scenario particularly interesting and original. Overall, I believe this work will attract broad interest at the interface of the photonic and Moiré physics communities. I therefore recommend publication in Nature Communications after the authors address the following points:

Response: We appreciate the reviewer's recognition of the importance and correctness of our work, and recommendation for publication in Nature Communications. We also thank the reviewer for the constructive comments to improve our work. In the following, we respond each individual comment in detail, along with corresponding changes made in the revised manuscript.

i) The authors correctly point out that the bilinear coupling term formally shares the same structure as the Moiré Hamiltonian, leading to an additional periodicity. However, for Moiré physics of real interest to emerge, the induced potential must be strong enough to produce flat bands, i.e., to substantially modify the band curvature, with miniband separations large compared to the exciton thermal distribution. Otherwise, excitons will experience a spatial modulation without true localization. In the present case, the folding vector q is small, so the ground state appears flat only because it is very close to the parabolic minimum, and already the third band in Fig. 2c shows a more pronounced dispersion. This raises the question: to what extent does localization actually occur, given that at low temperatures a thermal distribution spans ~ 20 meV, whereas the miniband separation in Fig. 2c is less than 10 meV?

Response: As the reviewer points out, at room temperature the thermal energy (≈ 25 meV) is larger than the minibands separation. However, at cryogenic temperatures (for instance liquid nitrogen, $T = 77$ K), the thermal energy is ≈ 6.6 meV, hence the separation in Fig. 2 can be resolved.

We stress that our focus is on the confinement of the excitons, not on their localization. The same goes

for the discussion of Fig. 2, and more generally in Section II.B. By excitonic confinement we mean that excitons feel a potential which comes from the interaction with the driving field, and the potential has the effect of folding the bands and splitting them at the border of the Moire BZ, as opposed to a free particle dispersion characteristic of unconfined (free) excitons. Both signatures can be seen in Fig. 2(c).

Conversely, by excitonic localization we mean that excitons are trapped in a potential well, whose signature is the presence of a flat dispersion. The miniband flatness is determined by the ratio between the strength of the interaction and the parabolic free excitonic dispersion $E_R = Q^2/(2M_{exc})$. Practically, using a longer-period grating (smaller q) and/or increasing $\tilde{A}_{0,q}$ (e.g., higher field enhancement or smaller mode volume) flattens the bands and enlarges the miniband gaps. Note that since this a Floquet driven cavity, it is always possible to increase the coupling by increasing the driving. However, as the reviewer points out, the BZ is small and over such a narrow range a parabolic band near its minimum can appear flat. For this reason we do not associate the seemingly flat branch to a localized phase.

Finally, let us consider the experimental feasibility of the parameters used for Fig. 2. The strength of interaction Hamiltonian responsible for the photon-induced band depends, as discussed in the Supplementary Info III, on $\tilde{A}_{0,q}\mathcal{B}_{ll',ij,Q,q}^{v,\lambda}$, where $\mathcal{B}_{ll',ij,Q,q}^{v,\lambda}$ depends linearly on q (c.f. Supplementary Info V.C) and $\tilde{A}_{0,q} = E_{0,q}/\omega_q$, where $E_{0,q}$ is the amplitude of the electric field. As we discuss in Section III of the main text, the value of the mode momentum q is determined by the periodicity of the grating. In particular, to achieve $q = [\pm 0.009, 0.0]$ a.u. one needs a grating with periodicity $d \approx 40$ nm, which should be accessible given the current state-of-the-art fabrication techniques. As for the coupling strength, typical values for Fabry-Perot cavities are $E_{0,q} \approx 10 \div 100$ kV/m and $E_{0,q} \approx 1 \div 10$ MV/m for sub-wavelength cavities, as it can be calculated from [Phys. Rev. Research 7, 033163]. The conversion rate for the coupling strength is $\tilde{A}_{0,q}$ [a.u.] = $\tilde{A}_{0,q}$ [V s / m]/ $1.24e-5$. Hence, to obtain $\tilde{A}_{0,q} = 0.04$ [a.u.] with $\omega_q = 1.5$ eV one needs $E_{0,q} > 10$ MV/m which is very challenging for the current cavity technology both with Fabry-Perot and sub-wavelength cavities under the vacuum fluctuating photon conditions. Conversely, to obtain $\tilde{A}_{0,q} = 0.02$ [a.u.] with $\omega_q = 0.05$ eV one needs $E_{0,q} \approx 3$ MV/m, which is feasible using a sub-wavelength cavity. Note that Fig. 2 refers to a driven cavity, hence the aforementioned conditions can be achieved by increasing the classical pump.

ii) Related to the above, if one increases the cavity parameters to enhance the strength of the bilinear term to overcome the point mentioned above, is it still valid to use the unmodified Wannier exciton states to define the exciton basis? In Moiré systems, the effective potential originates from atomic displacements and does not significantly alter the underlying Wannier exciton problem in first approximation, making it possible to treat it via an effective static continuum potential. By contrast, in the photonic case the periodic potential arises from light-matter interaction itself, and stronger coupling may renormalize the exciton wavefunctions directly. Strictly speaking, such a term should then be included in the Wannier equation to obtain new exciton states. I do not expect the authors to perform such a calculation, but I suggest that they comment on the range of validity of their approximation.

Response: In our formalism the light couples to the excitons via the bilinear coupling terms that, in the exciton representation, act on the center-of-mass degree of freedom (scattering $Q \rightarrow Q \pm q$). As

the reviewer points out, issues may arise when the coupling starts to significantly affect the excitonic wavefunction such that the exciton state itself must be recomputed.

One phenomenon that requires recomputing the excitonic wavefunction is the mixing of the $1s$ exciton with higher excitonic states (e.g. $2p$, $3d$). In fact, relaxing the truncation on the excitonic basis allows for internal wavefunctions rearrangements via mixing with the higher excited states (c.f. Nano Lett. 2019, 19, 6, 3473–3479). The theory we present in Section II.A also accounts for this effect, while in our calculation we used only the $1s$ state due to computational reasons. Here, we justify such numerical approximation. For Fig. 2, the binding energy of the intralayer exciton is $E_b \approx 0.368\text{eV}$, while the energy difference between the $1s$ and $2s$ state is $\Delta E_{b,1s-2s} \approx 0.193\text{eV}$. The mixing of the excitonic ground state with higher states is controlled by the ratio between the interaction energy and the energy difference between such states. If such ratio is significantly less than 1, then the approximation of considering only the $1s$ state has a small influence on the results. For the values used in Fig. 2, $\frac{\tilde{A}_{0,q} \mathcal{B}_{ll',ij,Q,q}^{v,\lambda}}{\Delta E_{b,1s-2s}} \approx 0.3$. Including the $2s$ exciton would therefore yield a minor improvement in accuracy that affects only numerical values, not the qualitative conclusions, while substantially increasing computational cost. We thus consider the present approximation justified for the scope of the main text.

One may also consider a spatial argument for using the unmodified Mott-Wannier equation. In the exciton relative-coordinate decomposition, the cavity field with in-plane wavevector \mathbf{q} varies across the exciton on the scale of the excitonic radius r_b . In the case of Fig. 2(c) such radius is $r_b \approx 1.64\text{nm}$, in agreement with previous works (Phys. Rev. B 92, 245123). Hence, the periodicity of the grating should be such that the field can be assumed constant over the excitonic radius r_b . For the values of \mathbf{q} used throughout this work we have estimated a real-space periodicity of the grating (hence of the field) which ranges from $d \approx 10\text{nm}$ to $d \approx 40\text{nm}$, c.f. Section III of the main text. Since these values are an order of magnitude bigger than the excitonic radius, we expect that using the unmodified Mott-Wannier equation is a valid approximation, provided that the $\mathbf{q} = 0$ excitonic problem for the equivalent coupling strength shows that the wavefunction is not altered. A grating periodicity of $d = 1.64\text{nm}$ would generate a mode momentum of $\mathbf{q} \approx [\pm 0.2, 0.0]\text{a.u.}$. It should also be noted that realizing a grating with such small periodicity is experimentally challenging, and unlikely to be seen in the near future.

Finally, one could also argue that even the screened interaction that is used to define the excitonic problem is affected by the coupling with the cavity. However, in order to account for that one would need to build the theory starting from light-dressed orbitals, for example using QEDFT, which is beyond scope for the present work.

We added the following paragraph in the Supplementary Information, Section V.A: *The use of unmodified Mott-Wannier exciton states assumes that the mixing between the $v = 1s$ excitonic state and higher excited states is small. This approximation is justified as long as the the ratio between the interaction energy and the energy difference between such states is significantly less than 1. Note that including the excitonic $2s$ implies a major increase in the computational cost. Moreover, we assume that the cavity field does not significantly distort the internal exciton wavefunction. Such an approximation is*

justified as long as long the light matter interaction does not distort the internal degrees of freedom of the exciton (i.e. the relative position of the electron and of the hole). Hence, the spatial variation of the cavity field, which is determined by the periodicity of the grating, must be small relative to the exciton size (i.e. to the excitonic radius), which is the case for the investigated periodicities.

iii) At the end of the first paragraph of the “Dark cavity” section, the authors cite only Ref. 14. I suggest also citing Nano Lett. 2022, 22, 11, 4468–4474, where exciton–polariton Moiré structures were computed. Although that work addresses a non-dark cavity, it provides a useful comparison and would strengthen the discussion.

We thank the reviewer for suggesting the work Nano Lett. 2022, 22, 11, 4468–4474. After reading it, we decided to include it as follows in the main text:

- In the introduction: *”We find that spatially unstructured cavities, where the electromagnetic field carries no momentum, can alter the twist induced excitonic confinement from the Moiré potential when the cavity-mode is resonant with the excitonic transition. This result is in agreement with previous works [Nano Lett. 2022, 22, 11, 4468–4474], where they applied a Hopfield diagonalization to a tight binding model to study how bare photonic and excitonic states contribute to exciton-polariton states in unstructured cavities at different twist angles. We show that the same result can be obtained using a more general theory, starting by a first principles QED Hamiltonian beyond the RWA approximation, and focusing on controlling the localization of exciton-polaritons.”*
- We added the citation at the point suggested by the reviewer
- In paragraph 2 of Section II.C: *”Interestingly, the UP in Fig.3(b-c) has the same dispersion as in the uncoupled matter system (Fig. 3(a)), whereas the LP line is almost constant for all twist angles, in agreement with previous works [Nano Lett. 2022, 22, 11, 4468–4474]. . . . Ref. [Nano Lett. 2022, 22, 11, 4468–4474] provides with an interesting and complementary discussion on how bare photonic and excitonic states contribute to the UP and LP;. In short, . . . ”*
- In the caption of Fig. 3: *”(a) Twisted heterostructure without the cavity (see Ref. [Nano Lett. 2020, 20, 12, 8534–8540] Fig. 5(a) and Ref. [Nano Lett. 2022, 22, 11, 4468–4474] Fig. 4(a)). (b, c) . . . Note that for these two panels Ω_c is significantly smaller than the excitonic resonance due to the diamagnetic shift. These results are qualitatively in agreement with Ref. [Nano Lett. 2022, 22, 11, 4468–4474] Fig. 4(b).”*

The authors might find the following list of minor corrections useful (some may be due to PDF conversion): Page 1, first column, first line: “Van der Waals” → “van der Waals”. Page 5, second column, last line: “form” → “from”. Use of hyphens: “hetero-structure” → “heterostructure”, “bi-layer” → “bilayer”, “bi-linear” → “bilinear”, “intra-layer” → “intralayer”. In the SI, Eqs. (17) and (19): the exciton annihilation operator is missing the index ν .

We thank the reviewer for spotting these typos. We fixed them in the manuscript.

2 Reply to Reviewer 2:

This work proposes and theoretically demonstrates an original approach for creating moiré systems without twists using structured planar optical cavities. The proposed schemes uses a TMD heterostructure embedded in a planar cavity with faces parallel to the structure. The authors show that, in the case of a structured cavity, the excitonic dispersion relations have characteristics of a moiré. The authors study the case of cavities driven by a classical field or without driving. In the second case, the interaction between the van der Waals heterostructure and the cavity induces long-range interactions.

This is an original approach, likely to open a new avenue in the field of moiré systems, avoiding twisting of solid-state structures. Moiré systems are attracting growing attention and the field is very active at the interface of condensed matter physics, quantum optics, and quantum gases. This new work enriches it and I am convinced that it will be of great interest to the community. The presentation of the theoretical model is clear and sound. Useful technical details of the calculations are well described in the supplementary material. In addition, the paper is clearly written and will be accessible even for non-specialists in the field.

I recommend the publication of the manuscript after the authors have considered the following points:

Response: We appreciate the reviewer's recognition of the originality, clearness and accessibility of our work. We also thank the reviewer recommending the publication in Nature Communications and for the constructive comments to improve our work. In the following, we respond each individual comment in detail, along with corresponding changes made in the revised manuscript.

In section II.A, it would be useful to say something about the method used in numerical calculations.

Response: After reconsidering Section II.A, we noticed that it was not explicitly stated that the matrix elements \mathcal{M} and \mathcal{E} contain information of the Mott-Wannier states. We made it clear by adding the following sentences (in italics): " $E_{g,II'}$ is the energy gap and $E_{b,II'}^v$ is the binding energy, which we obtain by solving the Mott-Wannier equation (c.f. Supplementary Information Sec. V.A). \mathcal{M} is the matrix element of the Moiré potential in the excitonic basis, describing the Moiré scattering of excitons with different momenta. Note that \mathcal{M} carries the information on the Mott-Wannier states (c.f. Supplementary Information Eq. A25).".

We then added more details regarding the procedure we followed in Section III. We prefer this location to Section II.A as it already contained the definition of the observables used in this work, and because in Section II.A the reader should focus on the model rather than on the methods used to solve it. Here's what we added: *In order to obtain the full exciton-polariton states, we start by solving the Mott-Wannier equation in the k -space (c.f. Supplementary Information Sec. V.A), which gives us access to the binding energy and the Mott-Wannier states. We considered only the 1s intralayer exciton in the MoSe₂ layer. The Mott-Wannier states are used to construct the excitonic creation and*

annihilation operators, as well as all the matrix elements appearing in Eq. 3. Refer to Supplementary Information Sec. I for the complete derivation. We represent the photonic modes using the Fock number states $\{|0\rangle, |1\rangle\}$.

It is written that the results "show that it gives confinement signatures similar to a twist induced Moiré potential" but the authors do not write precisely what makes the signature unambiguous. A discussion of these signatures would be very useful, particularly for the non-specialist reader.

Response: We thank the reviewer for raising this important point. We clarified in the main text what we mean by "confinement signatures" by adding the following (in italics): "... show that it gives confinement signatures similar to a twist induced Moiré potential, *which include the formation of a Moiré Brillouin Zone, and the opening of miniband gaps.*"

It is also written that "B shares the same excitonic operators of the Moiré potential M" but in a paragraph where it is written that $M=0$ a few lines above. This is very misleading.

Response: We thank the reviewer for pointing out this potential confusion. We decided to remove "(i.e. $M = 0$)" for that point, and we added it later as follows:

"Fig. 2 shows the results for a classically driven unstructured cavity as well as for a spatially structured cavity (with a one-dimensional spatial periodicity) both coupled to the $1s$ intralayer exciton in the MoSe_2 layer within the *untwisted heterostructure (i.e. $M = 0$).*"

A few lines below eq 2, please specify that z is the direction orthogonal to the cavity planes.

Response: We added the following bracket: "... are standing waves in the z direction (*which is orthogonal to the cavity planes*)".

The authors write "We assume that the matter couples with the in-plane component of the electric field and therefore only consider modes for which it is finite." What does justify this assumption? To what extent is this justified in practical implementation?

Response: We couple light and matter through minimal coupling, performing the canonical momentum substitution $\mathbf{p} \rightarrow \mathbf{p} + \mathbf{A}$ (in atomic units). Hence, light and matter are coupled through the bilinear coupling $\mathbf{p} \cdot \mathbf{A}$, which yield matrix elements which are proportional to $\mathbf{p} \cdot \mathbf{e}$, where \mathbf{e} is the polarization vector of the field. In a 2D material such the one considered in the main text, a particle has a significantly non-zero value of the momentum only in the in-plane direction (as it cannot move in the orthogonal one). Since $p_x, p_y \neq 0, p_z \approx 0$, the coupling happens only for the in-plane components.

This assumption is also well justified experimentally (Flatten et al., Sci. Rep. 6, 33134 (2016)). Planar microcavities used with 2D materials are operated in TE-like configurations with strong in-plane fields at the layer position, and the observed Rabi splittings in TMD polariton experiments arise from in-plane coupling. TM modes can have a finite E_z , but their overlap with the 2D current is small and their coupling to the bright excitons is negligible in typical geometries. Including a small out-of-plane

component would only introduce negligible corrections and would not affect our conclusions.

Page 6, right-hand-side column, it is written “In the range of small twist angles, however, we can identify localized polaritonic states [14].” What does induce localization? In fact, I am surprised that the localization appears at low angles but apparently not at larger angles, according to the text. I would expect the following phenomenology: For certain particular angles, the system is periodic and cannot induce localization; For most angles, on the other hand, the system is non-periodic and can induce localization. For very low angles but inducing a periodic structure, the moiré period is very large so that the couplings between subsequent cells are very weak and very small fluctuations can overcome them and give a false apparent localization. This was recently analyzed in a slightly different context in PRR 6, L042066 (2024) and PRA 111, 043305 (2025). Can the results observed in the manuscript be explained in a similar way?

Response: We thank the reviewer for this insightful comment. The highlighted sentence is indeed ambiguous. We defined localization according to Ref. 14 (S. Brem, Nano Letters 20, 8534 (2020)), which states that an excitonic state is localized if its corresponding branch in the linear response is below the unperturbed exciton ground state (which is a free particle) and if its shape is not parabolic. In our Fig. 3(a-c), the energy of such state would be slightly above 1.48eV. In other words, at large angles (when the Moiré period is small), the excitonic ground state (lowest branch in Fig. 3(a)) behaves as it was unperturbed by the potential, hence approaching the unperturbed exciton ground states. While at large angles only the ground state is below this level, at small angles other states appear below the free particle level, hence our sentence *In the range of small twist angles, however, we can identify localized polaritonic states [14]*. As for the other branches, at large angles their shape is parabolic, which is a signature of delocalization (S. Brem, Nano Letters 20, 8534 (2020)).

In the two interesting works suggested by the reviewer, the authors analyze the localization of Moiré states in terms of tunneling between adjacent Moiré cells, using a tight-binding approach for quasi-crystalline structures. We believe that such methods are complementary but outside our present scope. Nevertheless, we believe that reasoning in terms of tunneling probability from a Moiré BZ to another may be useful for a qualitative understanding even in the case of the present work.

After reviewing the discussion, we believe that the sentence highlighted by the reviewer is a stand-alone sentence which does not add much to the discussion on the the UP and LP polaritonic branches carried out in that paragraph. Therefore, in order to avoid confusion of the reader, we decided to remove it. Furthermore, we added a citation to one of the suggested papers at the end of Section II.C: “In short, the effect of the twisting angle can be modified by the electromagnetic field of a cavity, which influences the excitonic response by causing Rabi splitting of the resonant transition. This results in a different localization for the upper and lower polariton, *and may pave the way for new quantum phases [PRR 6, L042066 (2024)]*”.

List of Changes

1. Fixed terminology and typos across the manuscript:

- “Van der Waals” corrected to “van der Waals”.
- Uniform use of “heterostructure” (replacing “hetero-structure”).
- Uniform use of “bilayer” (replacing “bi-layer”).
- Uniform use of “bilinear” (replacing “bi-linear”).
- Uniform use of “intralayer”/“interlayer” (replacing “intra-layer”/“inter-layer”).

2. Added sections for compliance with editorial policies:

- Data availability.
- Code availability.
- Authors contributions.
- Competing interests.

3. Clarifications and additions in the Introduction:

- Added: ”This result is in agreement with previous works [Nano Lett. 2022, 22, 11, 4468–4474], where they applied a Hopfield diagonalization to a tight binding model to study how bare photonic and excitonic states contribute to exciton-polariton states in unstructured Fabry-Perot cavities at different twist angles. We show that the same result can be obtained using a more general theory, starting by a first principles QED Hamiltonian beyond the RWA approximation, and focusing on controlling the localization of exciton-polaritons”

4. Theory section (Sec. II.A) clarifications:

- Added the following parts to clarify how we obtain the Mott-Wannier states (in italics): ” $E_{g,l\ell}$ is the energy gap and $E_{b,l\ell}^v$ is the binding energy, *which we obtain by solving the Mott-Wannier equation (c.f. Supplementary Information Sec. V.A).* \mathcal{M} is the matrix element of the Moiré potential in the excitonic basis, describing the Moiré scattering of excitons with different momenta. *Note that \mathcal{M} carries the information on the Mott-Wannier states (c.f. Supplementary Information Eq. A25).*”
- Clarified geometry, added: “which is orthogonal to the cavity mirrors”.

- Cited both Refs. [37,38] for absorbing the diamagnetic term via a Bogoliubov transformation.

5. Classically driven cavity (Sec. II.B) clarifications and additions:

- Removed an early “(i.e. $\mathcal{M} = 0$)” to avoid confusion; reintroduced it later when describing Fig. 2.
- Defined “confinement signatures”: “which include the formation of a Moiré Brillouin Zone, and the opening of miniband gaps”.
- Corrected typo (“stemming from”).

6. Dark cavity (Sec. II.C) clarifications and additions:

- Added: “in agreement with previous works [Nano Lett. 2022, 22, 11, 4468–4474]”
- Added cross-references noting qualitative agreement with Nano Lett. 2022, 22, 4468–4474: “Ref. [Nano Lett. 2022, 22, 11, 4468–4474] provides with a complementary discussion on how bare photonic and excitonic states contribute to the UP and LP;”.
- Removed an ambiguous sentence on “In the range of small twist angles, however, we can identify localized polaritonic states [14].”;
- Added a pointer to PRR 6, L042066 (2024) at the end of Sec. II.C: “and may pave the way for new quantum phases [PRR 6, L042066 (2024)]”.

7. Methods (Sec. III) expansion and workflow details:

- Added a clear workflow: “In order to obtain the full exciton-polariton states, we start by solving the Mott-Wannier equation in the k -space (c.f. Supplementary Information Sec. V.A), which gives us access to the binding energy and the Mott-Wannier states. We considered only the $1s$ intralayer exciton in the MoSe₂ layer. The Mott-Wannier states are used to construct the excitonic creation and annihilation operators, as well as all the matrix elements appearing in Eq. 3. Refer to Supplementary Information Sec. I for the complete derivation. We represent the photonic modes using the Fock number states $\{|0\rangle, |1\rangle\}$.”

8. Figure and caption updates:

- Fig. 3 caption: added literature pointers for comparison (“see Ref. [14] Fig. 5(a) and Ref. [36] Fig. 4(a)”) and stated qualitative agreement with Ref. [36] Fig. 4(b).

9. Supplementary Information (SI) corrections and additions:

- Added a paragraph in SI Sec. V.A on the validity of using unmodified Mott–Wannier

exciton states: *The use of unmodified Mott-Wannier exciton states assumes that the mixing between the $\nu = 1s$ excitonic state and higher excited states is small. This approximation is justified as long as the ratio between the interaction energy and the energy difference between such states is significantly less than 1. Note that including the excitonic $2s$ implies a major increase in the computational cost. Moreover, we assume that the cavity field does not significantly distort the internal exciton wavefunction. Such an approximation is justified as long as the light-matter interaction does not distort the internal degrees of freedom of the exciton (i.e. the relative position of the electron and of the hole). Hence, the spatial variation of the cavity field, which is determined by the periodicity of the grating, must be small relative to the exciton size (i.e. to the excitonic radius), which is the case for the investigated periodicities.*

- Added missing index ν in the exciton annihilation operator in SI Section I, Eq. 17 and Eq. (19).

Responses to the reviewers comments

We sincerely thank the two reviewers for their efforts to carefully evaluate our manuscript. We feel that their comments have helped us improve our work and we are very pleased that both reviewers recognize its importance, significance and validity, and support its acceptance.

1 Reply to Reviewer 1:

The authors have thoroughly addressed all the comments raised in my previous review. The clarifications and revisions improve the presentation of the manuscript and provide sufficient detail regarding the methodology. I find the current version satisfactory and have no further comments. I therefore recommend the manuscript for publication in its present form.

Response: We sincerely thank the reviewer 1.

2 Reply to Reviewer 2:

The authors have carefully answered all my initial concerns and clarified all points. This is a very interesting and useful work. I fully support publication in the present form.

Response: We sincerely thank the reviewer 2.